# Preventing Overgrowth of Cucumber and Tomato Seedlings Using Difference between Day and Night Temperature in a Plant Factory with Artificial Lighting

**DOI:** 10.3390/plants12173164

**Published:** 2023-09-03

**Authors:** Young Ho Kim, Hwi Chan Yang, Yun Hyeong Bae, Soon Jae Hyeon, Seung Jae Hwang, Dea Hoon Kim, Dong Cheol Jang

**Affiliations:** 1Interdisciplinary Program in Smart Agriculture, Kangwon National University, Chuncheon 24341, Republic of Korea; rladydgh99@kangwon.ac.kr (Y.H.K.); jack955@kangwon.ac.kr (H.C.Y.); xhaahxj@kangwon.ac.kr (Y.H.B.); 2Department of Horticulture, College of Agriculture and Life Science, Kangwon National University, Chuncheon 24341, Republic of Korea; wo2091@naver.com; 3Department of Agricultural Plant Science, College of Agriculture & Life Sciences, Gyeongsang National University, Jinju 52828, Republic of Korea; hsj@gnu.ac.kr; 4Research Institute of Life Science, Gyeongsang National University, Jinju 52828, Republic of Korea; 5Institute of Agriculture & Life Sciences, Gyeongsang National University, Jinju 52828, Republic of Korea; 6Hoban Agriculture Corporation, Chuncheon 24211, Republic of Korea; jinhwaland@hanmail.net

**Keywords:** artificial lighting, overgrowth prevention, plant physiology, day and night temperature

## Abstract

This study aimed to determine the feasibility of temperature difference as an overgrowth-prevention technique to influence plant height and internode length in a plant factory with artificial lighting. The control plants were grown in a commercial nursery greenhouse using a growth regulator (Binnari), and +DIF (25 °C/15 °C), 0DIF (20 °C/20 °C), and −DIF (15 °C/25 °C) were the treatments with different day/night temperatures and the same average temperature (20 °C). Cucumbers showed the strongest suppression under the −DIF treatment, with a dwarfism rate of 33.3%. Similarly, tomatoes showed 0.8% and 22.2% inhibition in the 0DIF and −DIF treatments, respectively. The F_V_/F_M_ of cucumber was approximately 0.81 for all treatments. The OJIP changes differed for cucumbers; however, both cucumbers and tomatoes had similar OJIP curve patterns and no abnormalities. The relative growth rate of cucumbers at the growth stage was 1.48 cm·cm·day^−1^ for days 6–9 in +DIF stage 3, which was the highest growth rate among all treatments, and 0.71 cm·cm·day^−1^ for days 3–6 in −DIF stage 1, which was the most growth-inhibited treatment. In tomatoes, we found that days 3–6 of −DIF stage 1 had the most growth inhibition at 0.45 cm·cm·day^−1^. For cucumber, −DIF days 3–6 had the most growth inhibition, with a relative growth rate of 0.71 cm·cm·day^−1^, but the fidelity was significantly higher than the other treatments, with a 171% increase. The same was true for tomatoes, with days 3–6 of −DIF stage 1 showing the most inhibited growth at 0.45 cm·cm·day^−1^ but a 200% increase in fidelity. Therefore, applying the −DIF treatment at the beginning of growth would be most effective for both cucumbers and tomatoes to prevent overgrowth through the DIF in a plant factory with artificial lighting because it does not interfere with the seedling physiology and slows down the growth and development stage.

## 1. Introduction

Recent climate change and unusual weather have made growing high-quality, standardized seedlings a challenge because of issues such as growth inhibition due to high summer temperatures and extension of the seedling period during low winter temperatures, which affect physiology [1,2]. Therefore, plant factories with artificial lighting (PFALs), which are not affected by external weather conditions, are gaining attention as a technology for stable production of high-quality, standardized seedlings year round [3,4].

However, overgrowth due to the lack of light and high replanting density during the seedling process in PFALs is emerging [5,6]. To date, physical methods, such as contact stimulation, and chemical methods using growth regulators have been used to prevent overgrowth [7,8,9,10]. However, physical methods have disadvantages, such as initial installation costs, difficulty in implementation [11], and potential plant injuries. In the case of chemical methods, the positive list system, which was implemented in Korea from 1 January 2019, has been discouraged, along with conservation agriculture policies in every country, because of environmental pollution and human health effects [11,12].

Therefore, it is imperative to develop eco-friendly overgrowth prevention technologies to replace the existing overgrowth prevention methods. Recently, several studies have been conducted on overgrowth prevention methods, including wind flow [13], high-concentration potash fertilizer [14], ultraviolet-B (UV-B; 280–320 nm) [15], and the difference between day and night temperatures (DIF) [16,17,18,19,20]. Among these, the DIF method regulates growth according to the temperature difference between day and night. However, maintaining a nighttime temperature that is higher than the daytime temperature for a −DIF environment, where plant height and internode length are inhibited, is difficult in conventional greenhouses [21,22]. Plant factories, which are increasingly used, can use the DIF method to prevent overgrowth because the growing environment can be artificially controlled.

To date, most DIF studies have emphasized stem elongation or flower bud differentiation in flowering crops, such as roses, campanulas, salvias, kalanchoes, and lilies [17,19]. DIF-induced growth regulation studies on fruit and vegetable crops, such as bell peppers, cucumbers, and tomatoes, have also been conducted in conventional greenhouses but not in PFALs [23,24,25,26,27]. Furthermore, most studies have been conducted on +DIF methods, and there are very few studies on treatments at the nursery stage.

Hence, this study was conducted to check the feasibility of the DIF method, which affects plant height and internode length by controlling day and night temperatures, as a safe and eco-friendly overgrowth control method in PFALs that, unlike conventional methods, does not use physical or chemical techniques, such as growth regulators, that risk damaging the plant body. 

## 2. Results and Discussion

The effects of the DIF method on the aboveground morphological growth traits of cucumber and tomato seedlings were investigated (Table 1). The hypocotyl length of cucumbers was the longest in the control group (6.6 cm) and significantly shorter under the −DIF treatment (3.5 cm; approximately 26–89%) compared to other treatments. In tomatoes, it was the longest under the +DIF treatment (5.0 cm) and the shortest under the −DIF treatment (3.2 cm; approximately 16–56%), similar to cucumbers. The epicotyl was the longest under the +DIF treatment for both cucumbers and tomatoes at 8.2 and 2.5 cm, respectively. This concurred with the results published by Mius et al. [28], who found that tomato length growth increased more under a +DIF treatment than under a −DIF treatment. For the cucumber hypocotyl, the control group presented the shortest (1.4 cm); however, no statistically significant differences were noted under the −DIF treatment, and the tomato hypocotyl was the shortest at 1.5 cm under the −DIF treatment. In contrast to hypocotyl length, epicotyl length was reduced in the cucumber and tomato control treatments. This was likely due to the disruption of gibberellin biosynthesis in plants by the growth regulator diniconazole [29,30]. Cucumbers had the thickest stem diameter under the +DIF treatment (5.1 mm), whereas tomato stems were significantly thinner under the +DIF treatment (1.6 mm). SPAD value-based chlorophyll content was significantly lower in both cucumbers and tomatoes under the −DIF treatment (24.9 and 36.4, respectively), which was consistent with the findings described by Heins and Erwin [31], which suggested that chlorosis accelerates with decreasing chlorophyll content and increasing −DIF. These results were attributed to the decrease in carbohydrates in the leaves due to an increase in night respiration; however, chlorosis was caused by a decrease in photosynthetic efficiency due to low daytime temperature [32,33]. The leaf area of cucumbers was the highest under the +DIF treatment at 197.7 cm^2^ and the smallest in the control treatment at 62.3 cm^2^. However, tomatoes under the 0DIF treatment had the largest leaf area (21.9 cm^2^) and the control tomatoes had the smallest (14.1 cm^2^), which was consistent with the results obtained by Strang and Weis [34]. Yun et al. [35] also reported a decrease in the foliar area when tomato and strawberry plants were treated with the triazoles diniconazole and paclobutrazol. Regarding the dwarfism rate, cucumbers showed the greatest suppression in the −DIF treatment (33.3%). In tomatoes, a suppressive effect was observed on hypocotyl length in the 0DIF and −DIF treatments, with 0.8% and 22.2% reductions, respectively.

The aboveground and underground weight characteristics of cucumber and tomato plants were compared according to the DIF, as shown in Table 2. For the aboveground part, the fresh weight of cucumbers was the heaviest under the +DIF treatment (7.93 g), showing a large total plant height and leaf area, and the lightest under the control treatment (2.68 g). This was because diniconazole, a growth regulator, reduces herbage and leaf area. For the underground part, cucumbers under the −DIF treatment had the lightest weight (0.50 g). Tomatoes were the heaviest under the 0DIF and control treatments, weighing 0.79 and 0.76 g, respectively. The underground part was also the heaviest under the control treatment (0.18 g). For dry weight, both cucumbers and tomatoes showed similar trends as for fresh weight. These results were similar to those of Agrawal et al. [26]. Fresh and dry weights were lower under the −DIF treatment than under the +DIF treatment, and in the case of tomatoes, dry weight increased with increasing daytime temperature and DIF, similar to the results of Lim et al. [36]. For the shoot dry matter rate, cucumbers had the smallest fresh weight under the +DIF treatment of 7.9% and the highest under the −DIF treatment of 11.0%, which may have been due to the high respiration rate of assimilates, compared to those under other treatments because of high daytime temperature with insufficient light intensity inside the plant factory and large leaf area [37]. The control treatment resulted in the smallest shoot dry matter in tomatoes at 9.0%, and no statistically significant differences were noted among the remaining treatments.

The results for root length and diameter used to compare the rhizosphere morphological characteristics of cucumbers and tomatoes according to the DIF (Figure 1) showed that the +DIF treatment resulted in the longest total root length (248.4 cm), whereas the control treatment had the shortest (189.9 cm). However, for tomatoes, the control treatment resulted in the longest total root length (75.8 cm) and the +DIF treatment had the shortest (56.0 cm). The development of the total root surface area and average root diameter showed a pattern similar to that of the total root length. The underground part of the cucumbers was the most sluggish under the control treatment, which was consistent with the results of Kim et al. [11], who reported inferior growth of the underground part under the diniconazole treatment. These results were attributed to the diniconazole-led inhibition of gibberellin biosynthesis in cucumber seedlings and a decrease in underground part growth [11]. Cucumbers had significantly higher total root volume under the 0DIF treatment (0.58 cm^3^), and it was the smallest under the −DIF treatment (0.47 cm^3^). For tomatoes, a difference was found between the values; however, this difference was not statistically significant. The greater variation in root development with the DIF treatments in cucumber compared to tomato was likely because cucumber is more sensitive to DIF treatments at the nursery stage, as day and night temperatures independently have a greater effect than the DIF on tomato [38].

Chlorophyll fluorescence parameters (F_0_, F_V_, and F_M_) were measured to derive the maximum quantum yield of dark-adapted PSII (F_V_/F_M_), the electron transport rate through PSII (F_M_/F_0_), and OJIP curves (Table 3 and Figure 2). F_0_ and F_V_/F_M_ represent the activity of photosystem II and are used as indicators of plant responses to environmental stress [39]. F_0_ refers to the fluorescence emitted before the light energy in the plant body is transferred to the photosystem II reaction center, whereas F_V_/F_M_ represents the photochemical efficiency of photosystem II [40]. In general, F_0_ increases and F_V_/F_M_ and F_M_/F_0_ decrease when photosystem II is damaged; therefore, they are mainly used as stress indicators [41,42]. For healthy plant leaves, the F_V_/F_M_ ranges from 0.78 to 0.83, and for cucumber, it was approximately 0.81; all treatments were within the normal range [43,44]. This is because the stress caused by day–night temperature reversal under the DIF treatments did not directly damage photosystem II. Onoriodo et al. [45] showed that F_V_/F_M_ decreased as night temperature increased, which differed from the results of this experiment, where both cucumbers and tomatoes had the lowest F_V_/F_M_ under the +DIF treatment, in which night temperature was the lowest. We believe that this was due to the differences in responses between crops and the reversal of day/night temperatures in the −DIF treatment, as well as the increase in nighttime temperatures. Both cucumbers and tomatoes had the same low F_0_ values under the −DIF treatment, which was likely due to the lower chlorophyll content in the leaves in the DIF treatment (Table 1). The F_M_/F_0_ of both cucumbers and tomatoes showed the same trend as the F_V_/F_M_. The +DIF group had the lowest F_M_/F_0_. This appears to have been due to an increase in the minimum fluorescence yield (F_0_) rather than a decrease in the maximum fluorescence yield (F_M_). The chlorophyll fluorescence OJIP changes (OJIP curves) in cucumber and tomato plants were measured after the DIF treatment (Figure 2). In the case of cucumbers, differences were observed in the values in each section, but the OJIP curve pattern was similar and no abnormalities were noted. In the case of tomatoes, the value with the 0DIF treatment was elevated in the I-P section, but the curve patterns and values of the treatment sections, except for that of the 0DIF treatment, were almost similar, and no abnormalities were observed. This may have been because cucumbers are more affected by the DIF than tomatoes, which are independently affected by day and night temperatures [38]. Based on the changes in OJIP, it was concluded that there was a small difference in the photochemical reaction efficiency of photoperiod II under the DIF treatment [39].

Relative growth rates were used to compare the growth rates of cucumbers and tomatoes at different growth stages in response to day–night temperature differences, and the results are presented as box and swarm plots (Figure 3). The total relative growth rate of cucumbers was the highest under the +DIF treatment (average = 1.33 cm·cm·day^−1^), whereas the 0DIF treatment was similar to the +DIF treatment (average = 1.23 cm·cm·day^−1^), and the median (second quartile) difference between treatments was not significant. However, 75% (third quartile) of plants from the +DIF treatment were larger than those from the 0DIF treatment. Similarly to cucumbers, tomatoes had the highest total relative growth rate under the +DIF treatment (average = 1.49 cm·cm·day^−1^) and the lowest under the −DIF treatment (average = 1.39 cm·cm·day^−1^). The median (second quartile) for tomatoes was similar for the +DIF and −DIF treatments, with the 0DIF treatment presenting the highest. However, the area below 25% (first quartile) for the 0DIF and −DIF treatments was wider than that for the +DIF treatment. The relative growth rate of cucumbers according to growth stages was the highest among all treatments (1.48 cm·cm·day^−1^) in days 6–9 of the +DIF treatment (stage three), whereas the growth was most inhibited (0.71 cm·cm·day^−1^) in days 3–6 of the −DIF treatment (stage one). In tomatoes, days 6–9 of the 0DIF treatment (stage three) had the highest growth rate (1.42 cm·cm·day^−1^) and, similarly to cucumbers, days 3–6 of the −DIF treatment (stage one) had the highest growth inhibition (0.45 cm·cm·day^−1^). Hence, it was concluded that, to observe the effect of preventing overgrowth through DIF in cucumber and tomato seedlings in a plant factory with artificial lighting, treating both cucumbers and tomatoes at the early stage of growth, where the growth inhibition effect of the −DIF treatment was the greatest, would be effective.

To confirm the response of cucumber and tomato seedlings to DIF treatment, the growth stages were divided into early, mid, and late stages and examined at 3-day intervals, and the compactness of each growth stage is shown in Figure 4. Compactness is an indicator of seedling quality, with higher compactness indicating stronger seedlings [46]. In the +DIF treatment, the compactness of both cucumber and tomato seedlings increased at all growth stages as the number of treatment days increased. Cucumbers in stage one increased the most in days 6–9, with 15.7 mg·cm^−1^ (approximately 166%) and 11.8 mg·cm^−1^ (approximately 161%) under the +DIF and –DIF treatments, respectively. In stage two, they increased the most in days 6–9, with19.8 mg·cm^−1^ (approximately 176%) under the 0DIF treatment. All treatments showed the smallest increase from day 6 to day 9 in stage three, with the +DIF treatment showing 9.8 mg·cm^−1^ (approximately 121%), the 0DIF treatment showing 17.3 mg·cm^−1^ (approximately 137%), and the −DIF treatment showing 19.6 mg·cm^−1^ (approximately 144%). For tomatoes, all treatments showed the largest increase in stage two on day 9 compared to day 6 at 6.7 mg·cm^−1^ (approximately 479%), 6.4 mg·cm^−1^ (approximately 316%), and 6.4 mg·cm^−1^ (approximately 317%). For the +DIF and −DIF treatments, similarly to cucumbers, stage 3 at day 9 compared to day 6 showed the smallest increases at 2.1 mg·cm^−1^ (approximately 117%) and 1.9 mg·cm^−1^ (approximately 113%). For cucumbers, days 6–3 of the –DIF treatment had the most growth inhibition, with a relative growth rate of 0.71 cm·cm·day^−1^, but the 171% increase in compactness was significantly higher than the other treatments. Similarly, for tomatoes, days 6–3 of the stage one of the −DIF treatment had the slowest growth with 0.45 cm·cm·day^−1^, but a 200% increase in compactness was observed (Figure 4). Combining the results for cucumber and tomato seedlings, we concluded that the −DIF treatment affected the growth and developmental stages rather than causing physiological disorders in the seedlings.

### 2.1. Plant Factory with Artificial Lighting Specifications

The PFAL was located as part of Hoban Agriculture Corporation (latitude 37°55’29″, longitude 127°47’04″, 85 m above sea level) (GMP Co., Ltd., Hwaseong, Republic of Korea). The exterior walls were insulated with urethane foam (70 mm) to provide protection from the external environment. The interior was equipped with an air conditioning system, seedling modules, a nutrient supply system, and an environmental control program. There were six nursery modules, each with five layers of Styrofoam beds. In each nursery module, five 28 W white LEDs were placed on the upper wall of the bed, and a control program was installed to control the light intensity, photoperiod, temperature, humidity, and watering. A unit cooler was installed on the upper wall at the center of the plant interior and used as an air-conditioning system for temperature control and internal air circulation (Figure 5).

### 2.2. Plant Materials and Growth Conditions

The cucumber and tomato varieties used in the experiment were Hangangmat (*Cucumis sativus* L.; Farm Hannong Co., Ltd., Seoul, Republic of Korea) and TY205 (PPS Co., Ltd., Yongin, Republic of Korea), respectively. The cucumber was sown on 28 September 2022 and entered into the factory on 2 October 2022, whereas the tomato was sown on 20 September 2022 and entered into the factory on 23 September 2022. Horticultural soil (electrical conductivity (EC): 0.47 dS·m^−1^, pH 6.18; Pindstrup, Denmark) was filled into 162-hole trays (W 280 × L 540 × H 45 mm^3^; Bumnong Co., Ltd., Jeongeup, Republic of Korea) and seeds were sown. The sown trays were well irrigated with overhead irrigation and then germinated for 48 h in a dark germination chamber maintained at 25 °C–28 °C and 90% relative humidity.

After germination, plants were nursed in the PFAL. The environment in the PFAL was set to a 12 h/12 h photoperiod (day/night), and the light source was a white light-emitting diode (LED) with a photosynthetic photon flux (PPF) of 350 μmol·m^−2^·s^−1^. The relative humidity was 60%/70% (day/night) during the growth period. Irrigation was supplemental every three days at a pH of 5.5 and an EC of 1.4–1.45 dS·m^−1^ using a nutrient solution of Technigro 13-2-13 plus fertilizer (Sun-Gro Horticulture, Bellevue, WA, USA).

### 2.3. DIF Treatment of Cucumber and Tomato Seedlings

For the control, the growth regulator Binnari (diniconazole 5%; Dongbangagro Co., Ltd., Seoul, Republic of Korea) was applied to cucumber plants with fully developed cotyledons at 1.5 g/20 L (effective dose = 3.75 mg·L^−1^), and tomato was foliar-sprayed at 3 g/20 L (7.5 mg·L^−1^) after the emergence of three main leaves. The DIF treatments were +DIF (25 °C/15 °C), 0DIF (20 °C/20 °C), and −DIF (15 °C/25 °C), with different day/night temperatures and the same average temperature (20 °C) (Figure 6). As the seedling growth periods of tomato and cucumber are different, the treatment period differed for each crop according to the seedling growth period: 15 days after sowing (DAS) for cucumber and 19 DAS for tomatoes.

### 2.4. Investigation

The investigation of cucumbers and tomatoes was conducted by randomly sampling 30 plants for each crop when deviations in growth occurred due to DIF. To confirm the sensitivity of the growth stage to day/night temperature deviations, the growth stage was divided into early, middle, and late stages, and 20 experiments were conducted three times each at 3-day intervals for 60 investigations. The investigations included investigations of hypocotyl length, epicotyl length, stem diameter, leaf area (LI-3100; LI−COR Inc., Lincoln, NE, USA), SPAD, shoot fresh weight, and dry weight. The dry weight was measured after 72 h of hot air drying (convection oven, SANYO Inc., Osaka, Japan) at 80 °C after measuring fresh weight. In the rhizosphere, root length, root diameter, total surface area, and total volume were measured using a WinRHIZO (WinRHIZO PRO 09; REGENT Instruments Inc., Quebec, QC, Canada). The following formulas for the dwarf ratio, relative growth rate, and dry matter rate were used based on the analysis criteria of the Agricultural Science and Technology Research Center of the Rural Development Administration [47].
$$\mathrm{Dwarf}\;\mathrm{rate}\;\left(\%\right)\;=\;\frac{(\mathrm{Control}\;\mathrm{plant}\;\mathrm{height}\;\left(\mathrm{cm}\right)\;-\;\mathrm{Treatment}\;\mathrm{plant}\;\mathrm{height}\;\left(\mathrm{cm}\right))}{\mathrm{Control}\;\mathrm{plant}\;\mathrm{height}\;(\mathrm{cm})}$$
$$\mathrm{Relative}\;\mathrm{growth}\;\mathrm{rate}\;\mathrm{of}\;\mathrm{plant}\;\mathrm{height}\;\left[{\mathrm{RGR}}_{H}\right](\mathrm{cm}\cdot \mathrm{cm}\cdot {\mathrm{day}}^{-1})$$
$$H_{0}\;\mathrm{and}\;H_{1}:\;\mathrm{initial}\;\mathrm{and}\;\mathrm{final}\;\mathrm{plant}\;\mathrm{height}$$
$$t_{1}\;-\;t_{0}:\;\mathrm{growing}\;\mathrm{period}\;(\mathrm{days})$$
$$\mathrm{Dry}\;\mathrm{matter}\;\mathrm{rate}\;\left(\%\right)\;=\;\frac{\mathrm{Fresh}\;\mathrm{weight}\;(g)}{\mathrm{Dry}\;\mathrm{weight}\;(g)}$$
$$\mathrm{Compactness}\;\left[\mathrm{CP}\right]\left(\mathrm{mg}\cdot {\mathrm{cm}}^{-1}\right)=\frac{\mathrm{Shoot}\;\mathrm{dry}\;\mathrm{weight}\;(\mathrm{mg})}{\mathrm{Plant}\;\mathrm{height}\;(\mathrm{cm})}$$

### 2.5. OJIP Measurement Methods and Parameters

A chlorophyll fluorescence analyzer (Fluorpen FP-110; Photon Systems Instruments, Drásov, Czech Republic) was used to determine the stress index of the seedlings in response to the DIF treatment. The first main leaves were randomly selected five times per treatment and measured after 15 min of adaptation to the dark. After the OJIP measurements, the OJIP indices F_0_, F_j_, F_i_, F_M_, F_M_/F_0_, and F_V_/F_M_ were calculated using the FluorPen program (Version 1.1.2.3; Photon Systems Instruments, Drásov, Czech Republic), and each index was tabulated [48,49] (Table 4).

### 2.6. Statistical Analysis

All the statistical analyses were performed using SPSS (version 26; IBM Corp., Armonk, NY, USA). Two-way ANOVA was conducted to test the interaction between two independent variables: crop (independent variable one) and treatment (independent variable two). To investigate the growth and development of cucumber and tomato seedlings in response to DIF treatment, the control and three DIF treatments were statistically tested for significance (*p* < 0.05) using Duncan’s multiple range test.

## 3. Conclusions

The application of DIF to prevent the overgrowth of cucumber and tomato seedlings in a PFAL resulted in growth inhibition in both the cucumber and tomato compared to the control with growth regulators in the −DIF treatment. For root development, cucumber was more sensitive to the DIF treatment compared to tomato, but there was no significant difference between treatments. For chlorophyll fluorescence, it was found that the −DIF treatment reduced the minimum fluorescence yield owing to chlorophyll content due to yellowing. However, little difference was observed in the photochemical reaction efficiency during photoperiod II, and the stress caused by the DIF treatment did not directly damage the photoperiod II system. Through this, no physiological disturbance was caused by the DIF treatment, and since it slowed down the growth and development stage, it was concluded that it was most effective to apply the −DIF treatment at the beginning of growth when the growth inhibition effect caused by the −DIF was high in both cucumbers and tomatoes. It was confirmed that the growth of both cucumber and tomato was inhibited when subjected to the −DIF treatment, but the degree of inhibition was severe, so there is a concern about effects on operating costs due to the extension of the seedling period.

Although application of DIF inhibits growth, further research is needed on the optimum temperature setting that would not affect the seedling period and on the effects of DIF treatment during the seedling period on post-dietary growth, as they are unknown.

## Figures and Tables

**Figure 1 plants-12-03164-f001:**
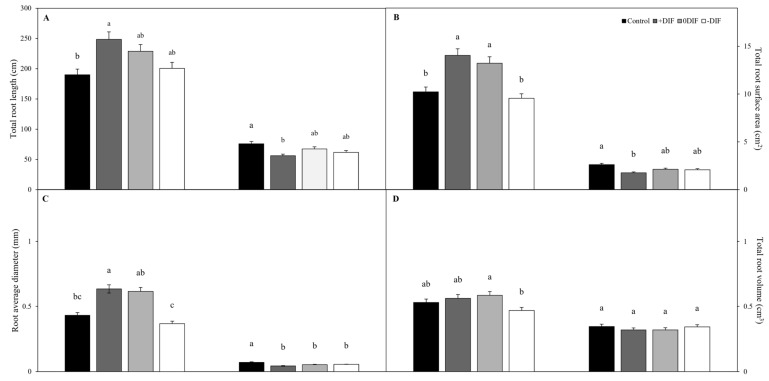
Root length (**A**), total root surface area (**B**), root average diameter (**C**), and total root volume (**D**) of cucumber and tomato in relation to the difference between day and night temperature. Significant differences between data are shown by different letters, *p* ≤ 0.05.

**Figure 2 plants-12-03164-f002:**
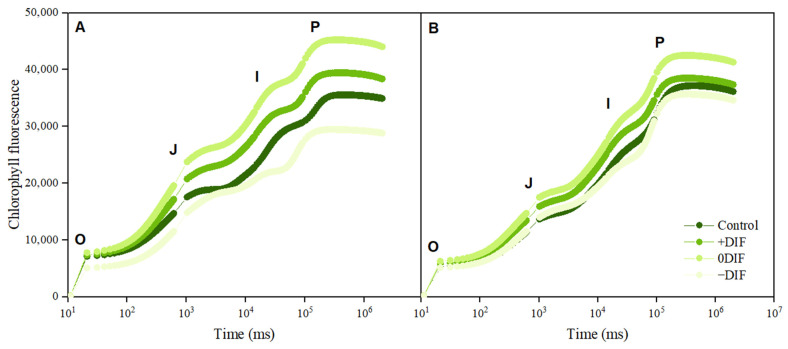
OJIP curves for cucumber (**A**) and tomato (**B**) grown under condition with a difference between day temperature and night temperature. +DIF: 25/15 °C (day/night); 0DIF 20/20 °C (day/night); −DIF 15/25 °C (day/night).

**Figure 3 plants-12-03164-f003:**
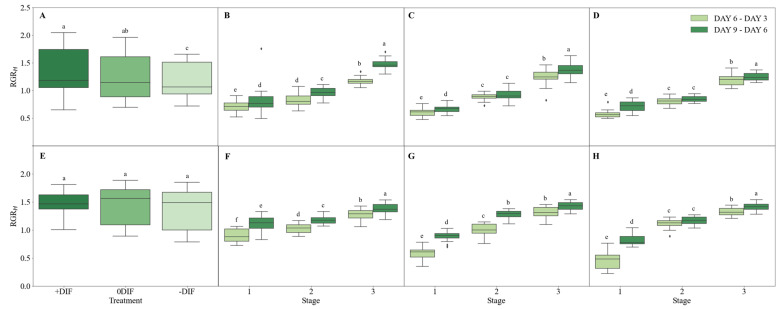
Growth stages of cucumber and tomato seedlings in DIF treatments and relative growth in plant height by day of treatment. (**A**) Relative growth rate of cucumber in 9 days before grafting; (**B**) relative growth rate of cucumber with +DIF treatment; (**C**) relative growth rate of cucumber with 0DIF treatment; (**D**) relative growth rate of cucumbers with −DIF treatment; (**E**) relative growth rate of tomato in 9 days before grafting; (**F**) relative growth rate of tomato with +DIF treatment; (**G**) relative growth rate of tomato with 0DIF treatment; (**H**) relative growth rate of tomato with −DIF treatment. Significant differences between data are shown by different letters, *p* ≤ 0.05.

**Figure 4 plants-12-03164-f004:**
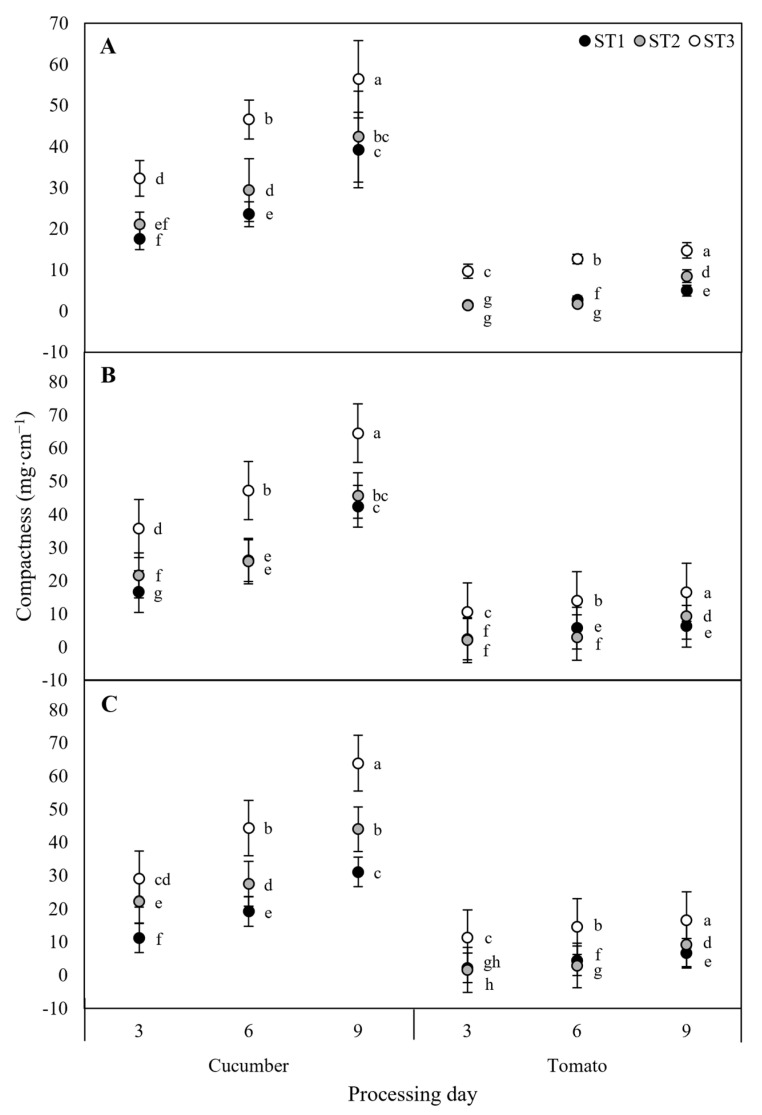
Growth stages of cucumber and tomato seedlings in DIF treatments and compactness by day of treatment. (**A**) +DIF treatment; (**B**) 0DIF treatment; (**C**) −DIF treatment. Significant differences between data are shown by different letters, *p* ≤ 0.05.

**Figure 5 plants-12-03164-f005:**
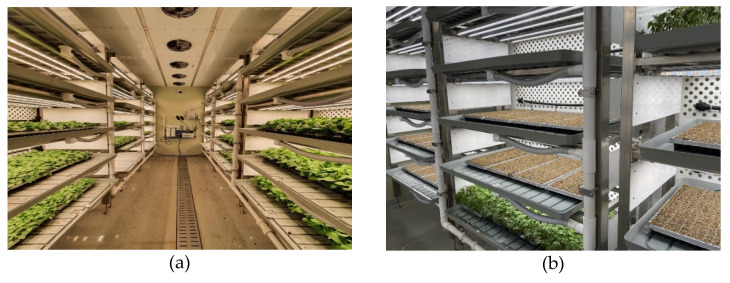
Interior of the experimental plant factory with artificial lighting used in research. Growing cucumber (**a**) and tomato (**b**) using PFAL.

**Figure 6 plants-12-03164-f006:**
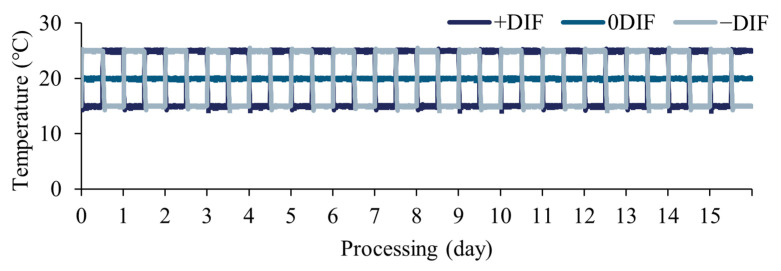
The day and night temperatures inside the plant factory with artificial lighting used in the study.

**Table 1 plants-12-03164-t001:** Growth characteristics of cucumber and tomato as affected by the difference between day temperature and night temperature conditions in a plant factory with artificial lighting.

Crop	Treatment (°C)	Hypocotyl Length (cm)	Epicotyl Length (cm)	Stem Diameter (mm)	Leaf Chlorophyll (SPAD)	Leaf Area (cm^2^)	Dwarf Rate ^z^ (%)
Cucumber	Con	6.6 ± 0.7 a ^y^	1.4 ± 0.3 c	3.0 ± 0.2 d	33.6 ± 3.6 b	62.3 ± 9.9 d	0 b
	25/15	5.1 ± 0.9 b	8.2 ± 1.9 a	5.1 ± 0.3 a	39.8 ± 3.6 a	197.7 ± 19.2 a	−66.7 ± 31.6 d
	20/20	4.4 ± 0.7 c	5.6 ± 1.4 b	4.4 ± 0.3 b	34.8 ± 4.7 b	171.8 ± 13.5 b	−24.7 ± 24.4 c
	15/25	3.5 ± 0.5 d	1.8 ± 0.4 c	3.5 ± 0.3 c	24.9 ± 2.7 c	117.5 ± 13.8 c	33.3 ± 8.7 a
Tomato	Con	4.3 ± 0.3 b	1.7 ± 0.3 c	1.8 ± 0.2 a	47.1 ± 3.2 a	14.1 ± 2.1 b	0 b
	25/15	5.0 ± 0.5 a	2.5 ± 0.4 a	1.6 ± 0.2 b	40.1 ± 4.4 b	14.7 ± 2.7 b	−24.1 ± 10.9 c
	20/20	3.7 ± 0.6 c	2.3 ± 0.5 b	1.8 ± 0.2 a	38.6 ± 3.6 b	21.9 ± 5.1 a	0.8 ± 11.5 b
	15/25	3.2 ± 0.2 d	1.5 ± 0.3 d	1.8 ± 0.2 a	36.4 ± 3.3 c	15.6 ± 3.2 b	22.2 ± 5.9 a
Significance ^x^	Crop (A)	***	***	***	***	***	***
	Treatment (B)	***	***	***	***	***	***
	A × B	***	***	***	***	***	***

^z^ Dwarf rate = (control plant height (cm) − treatment plant height (cm))/control plant height (cm) × 100. ^y^ Means with the same letters were not significantly different according to Duncan’s multiple range test (DMRT) at *p* ≤ 0.05. ^x^ *** indicate significance at *p* ≤ 0.001.

**Table 2 plants-12-03164-t002:** Weight characteristics of cucumber and tomato as affected by the difference between day temperature and night temperature conditions in a plant factory with artificial lighting.

Crop	Treatment (°C)	Fresh Weight (g)		Dry Weight (g)		Dry Matter Rate ^z^ (%)	
		Shoot	Root	Shoot	Root	Shoot	Root
Cucumber	Con	2.68 ± 0.39 d ^y^	0.60 ± 0.11 bc	0.230 ± 0.036 c	0.025 ± 0.004 b	8.6 ± 0.8 c	4.2 ± 0.3 b
	25/15	7.93 ± 0.87 a	0.81 ± 0.18 ab	0.629 ± 0.113 a	0.034 ± 0.009 a	7.9 ± 1.1 d	4.2 ± 0.4 b
	20/20	6.44 ± 0.61 b	0.87 ± 0.41 a	0.611 ± 0.097 a	0.032 ± 0.007 a	9.5 ± 1.5 b	4.1 ± 1.0 b
	15/25	3.99 ± 0.55 c	0.50 ± 0.17 c	0.444 ± 0.101 b	0.023 ± 0.006 b	11.0 ± 1.5 a	4.8 ± 0.6 a
Tomato	Con	0.76 ± 0.14 a	0.18 ± 0.04 a	0.069 ± 0.012 b	0.010 ± 0.002 a	9.0 ± 0.8 b	6.0 ± 2.5 b
	25/15	0.59 ± 0.11 b	0.10 ± 0.04 b	0.062 ± 0.014 b	0.007 ± 0.001 b	10.5 ± 1.0 a	8.2 ± 2.0 a
	20/20	0.79 ± 0.19 a	0.10 ± 0.04 b	0.079 ± 0.021 a	0.006 ± 0.002 b	10.1 ± 1.2 a	6.6 ± 2.2 b
	15/25	0.60 ± 0.12 b	0.12 ± 0.04 b	0.061 ± 0.013 b	0.007 ± 0.002 b	10.3 ± 2.3 a	6.2 ± 1.3 b
Significance ^x^	Crop (A)	***	***	***	***	***	***
	Treatment (B)	***	***	***	***	***	***
	A × B	***	***	***	***	***	***

^z^ Dry matter rate = fresh weight (g)/dry weight (g). ^y^ Means with the same letters were not significantly according to Duncan’s multiple range test (DMRT) at *p* ≤ 0.05. ^x^ *** indicate significance at *p* ≤ 0.001.

**Table 3 plants-12-03164-t003:** Chlorophyll fluorescence in cucumber and tomato grown under conditions with a difference between day and night temperature. +DIF: 25/15 °C (day/night); 0DIF 20/20 °C (day/night); −DIF 15/25 °C (day/night).

Crop	Treatment	F_0_	F_M_	F_v_	F_v_/F_M_ ^z^	F_M_/F_0_
Cucumber	Control	7192.5 ± 322.3 b ^y^	35,388.0 ± 951.9 b	28,195.5 ± 662.6 bc	0.797 ± 0.005 a	4.92 ± 0.12 a
	+DIF	7526.0 ± 517.9 ab	39,264.8 ± 4776.8 b	31,738.8 ± 4909.5 b	0.806 ± 0.033 a	5.24 ± 0.76 a
	0DIF	7972.8 ± 401.8 a	45,076.0 ± 3950.4 a	37,103.3 ± 4115.4 a	0.822 ± 0.020 a	5.67 ± 0.67 a
	−DIF	5169.5 ± 156.5 c	29,364.3 ± 1426.6 c	24,194.8 ± 1347.1 c	0.824 ± 0.008 a	5.68 ± 0.25 a
Tomato	Control	5256.5 ± 209.6 c	37,061.3 ± 1041.3 bc	31,804.8 ± 840.7 b	0.858 ± 0.002 a	7.05 ± 0.10 a
	+DIF	5901.5 ± 264.3 b	38,334.3 ± 2051.4 b	32,432.8 ± 1801.4 b	0.846 ± 0.003 c	6.49 ± 0.11 c
	0DIF	6371.8 ± 200.6 a	42,337.3 ± 1619.3 a	35,965.5 ± 1424.1 a	0.850 ± 0.002 bc	6.64 ± 0.07 bc
	−DIF	5229.5 ± 220.1 c	35,582.8 ± 1069.7 c	30,353.3 ± 949.8 b	0.853 ± 0.005 b	6.81 ± 0.23 b
Significance ^x^	Crop (A)	***	NS	*	***	***
	Treatment (B)	***	***	***	NS	NS
	A × B	***	**	*	NS	*

^z^ F_V_/F_M_ = (F_M_ − F_0_)F_M_. ^y^ Means with the same letters were not significantly different according to Duncan’s multiple range test (DMRT) at *p* ≤ 0.05. ^x^ NS: non-significant; *, **, and *** indicate significance at *p* ≤ 0.05, 0.01, and 0.001.

**Table 4 plants-12-03164-t004:** Definitions of parameters obtained from the recorded chlorophyll fluorescence origin jump intermediate peak (OJIP) transients.

Parameter	Equation	Definition
F_0_		Minimal fluorescence yield of dark-adapted PSII
F_j_		Fluorescence intensity at J-step (at 2 ms)
F_i_		Fluorescence intensity at I-step (at 60 ms)
F_M_		Maximal fluorescence yield of dark-adapted PSII
F_M_/F_0_		Electron transport rate through PSII
F_V_/F_M_	F_V_/F_M_ = (F_M_ − F_0_)/F_M_	Maximum quantum yield of dark-adapted PSII

## Data Availability

Data are available in a publicly accessible repository.

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
