# Peer review of "Preventing Overgrowth of Cucumber and Tomato Seedlings Using Difference between Day and Night Temperature in a Plant Factory with Artificial Lighting"

_plants, 2023, doi:10.3390/plants12173164_

Round 1
Reviewer 1 Report
ï‚· There is mix between the results of Hypocotyl and Epicotyl (lines 83-85)
ï‚· Line 99 stated “Similarly, for tomatoes, 0DIF had the largest leaf area (21.9 cm2 )……” this is not right as for Cucumber +DIF has the largest leaf area, so it Is not similarly.
ï‚· I have missed the day length in the experiment description . the focus is on Temperature but that is in combination with the day length!
ï‚· The research has been conducted using single cucumber plants (genotype) and single tomato plants (genotype): however there is always genotype dependency within the species. I wounder why not more plants/genotypes have been used for each of the two studied species!
ï‚· The light intensity is limited to one condition, what is the impact of light intensity vs. Temperature for controlling and steering the growth in plant factory? That will limit the implantation of the results in different plant factories.
ï‚· There is no table one? The result starts with table 2!
ï‚· Many measurements have been done (presented in table 2, 3, 4….) however the conclusions were built only on chlorophyII results! What is the relevance of all the measures and what is the determination measurement that reflect/indicate overgrowth prevention? The conclusions can be extended to reflect on this aspect in my opinion.
Reviewer 2 Report
In this work, the authors investigated the effect of the difference between day and night temperatures on the growth of cucumber and tomato seedlings. Materials and methods, results and discussions are described in detail.
The article needs a little revision.
1. In the introduction, an overview of recent research on this topic should be expanded.
2. Add keywords.
3. At the end of the introduction, emphasize the novelty of your work. Lack of justification why this study needs to be carried.
4. Check text formatting.
5.The inscriptions in Figure 5 are hard to read.
6. What was the intensity of artificial lighting (PPFD)?
7. The results obtained look obvious and predictable, it is necessary to highlight the novelty in the conclusions section.
